# Gradient Descent Can Take Exponential Time to Escape Saddle Points

**Simon S. Du**
Carnegie Mellon University
ssdu@cs.cmu.edu

**Chi Jin**
University of California, Berkeley
chijin@berkeley.edu

**Jason D. Lee**
University of Southern California
jasonlee@marshall.usc.edu

**Michael I. Jordan**
University of California, Berkeley
jordan@cs.berkeley.edu

**Barnabás Póczos**
Carnegie Mellon University
bapoczos@cs.cmu.edu

**Aarti Singh**
Carnegie Mellon University
aartisingh@cmu.edu

## Abstract

Although gradient descent (GD) almost always escapes saddle points asymptotically [Lee et al., 2016], this paper shows that even with fairly natural random initialization schemes and non-pathological functions, GD can be significantly slowed down by saddle points, taking exponential time to escape. On the other hand, gradient descent with perturbations [Ge et al., 2015, Jin et al., 2017] is not slowed down by saddle points—it can find an approximate local minimizer in polynomial time. This result implies that GD is inherently slower than perturbed GD, and justifies the importance of adding perturbations for efficient non-convex optimization. While our focus is theoretical, we also present experiments that illustrate our theoretical findings.

## 1  Introduction

Gradient Descent (GD) and its myriad variants provide the core optimization methodology in machine learning problems. Given a function $f(\mathbf{x})$, the basic GD method can be written as:

$$\mathbf{x}^{(t+1)} \leftarrow \mathbf{x}^{(t)} - \eta \nabla f\big(\mathbf{x}^{(t)}\big), \tag{1}$$

where $\eta$ is a step size, assumed fixed in the current paper. While precise characterizations of the rate of convergence GD are available for convex problems, there is far less understanding of GD for non-convex problems. Indeed, for general non-convex problems, GD is only known to find a stationary point (i.e., a point where the gradient equals zero) in polynomial time [Nesterov, 2013].

A stationary point can be a local minimizer, saddle point, or local maximizer. In recent years, there has been an increasing focus on conditions under which it is possible to escape saddle points (more specifically, *strict* saddle points as in Definition 2.4) and converge to a local minimizer. Moreover, stronger statements can be made when the following two key properties hold: 1) all local minima are global minima, and 2) all saddle points are strict. These properties hold for a variety of machine learning problems, including tensor decomposition [Ge et al., 2015], dictionary learning [Sun et al., 2017], phase retrieval [Sun et al., 2016], matrix sensing [Bhojanapalli et al., 2016, Park et al., 2017], matrix completion [Ge et al., 2016, 2017], and matrix factorization [Li et al., 2016]. For these

problems, any algorithm that is capable of escaping strict saddle points will converge to a global minimizer from an arbitrary initialization point.

Recent work has analyzed variations of GD that include stochastic perturbations. It has been shown that when perturbations are incorporated into GD at each step the resulting algorithm can escape strict saddle points in polynomial time [Ge et al., 2015]. It has also been shown that episodic perturbations suffice; in particular, Jin et al. [2017] analyzed an algorithm that occasionally adds a perturbation to GD (see Algorithm 1), and proved that not only does the algorithm escape saddle points in polynomial time, but additionally the number of iterations to escape saddle points is nearly dimension-independent[1]. These papers in essence provide sufficient conditions under which a variant of GD has favorable convergence properties for non-convex functions. This leaves open the question as to whether such perturbations are in fact necessary. If not, we might prefer to avoid the perturbations if possible, as they involve additional hyper-parameters. The current understanding of gradient descent is silent on this issue. The major existing result is provided by Lee et al. [2016], who show that gradient descent, with any reasonable random initialization, will always escape strict saddle points *eventually*—but without any guarantee on the number of steps required. This motivates the following question:

**Does randomly initialized gradient descent generally escape saddle points in polynomial time?**

In this paper, perhaps surprisingly, we give a strong *negative* answer to this question. We show that even under a fairly natural initialization scheme (e.g., uniform initialization over a unit cube, or Gaussian initialization) and for non-pathological functions satisfying smoothness properties considered in previous work, GD can take exponentially long time to escape saddle points and reach local minima, while perturbed GD (Algorithm 1) only needs polynomial time. This result shows that GD is fundamentally slower in escaping saddle points than its perturbed variant, and justifies the necessity of adding perturbations for efficient non-convex optimization.

The counter-example that supports this conclusion is a smooth function defined on $\mathbb{R}^d$, where GD with random initialization will visit the vicinity of $d$ saddle points before reaching a local minimum. While perturbed GD takes a constant amount of time to escape each saddle point, GD will get closer and closer to the saddle points it encounters later, and thus take an increasing amount of time to escape. Eventually, GD requires time that is exponential in the number of saddle points it needs to escape, thus $e^{\Omega(d)}$ steps.

## 1.1 Related Work

Over the past few years, there have been many problem-dependent convergence analyses of non-convex optimization problems. One line of work shows that with smart initialization that is assumed to yield a coarse estimate lying inside a neighborhood of a local minimum, local search algorithms such as gradient descent or alternating minimization enjoy fast local convergence; see, e.g., [Netrapalli et al., 2013, Du et al., 2017, Hardt, 2014, Candes et al., 2015, Sun and Luo, 2016, Bhojanapalli et al., 2016, Yi et al., 2016, Zhang et al., 2017]. On the other hand, Jain et al. [2017] show that gradient descent can stay away from saddle points, and provide global convergence rates for matrix square-root problems, even without smart initialization. Although these results give relatively strong guarantees in terms of rate, their analyses are heavily tailored to specific problems and it is unclear how to generalize them to a wider class of non-convex functions.

For general non-convex problems, the study of optimization algorithms converging to minimizers dates back to the study of Morse theory and continuous dynamical systems ([Palis and De Melo, 2012, Yin and Kushner, 2003]); a classical result states that gradient flow with random initialization always converges to a minimizer. For stochastic gradient, this was shown by Pemantle [1990], although without explicit running time guarantees. Lee et al. [2016] established that randomly initialized gradient descent with a fixed stepsize also converges to minimizers almost surely. However, these results are all asymptotic in nature and it is unclear how they might be extended to deliver explicit convergence rates. Moreover, it is unclear whether polynomial convergence rates can be obtained for these methods.

Next, we review algorithms that can provably find approximate local minimizers in polynomial time. The classical cubic-regularization [Nesterov and Polyak, 2006] and trust region [Curtis et al., 2014]

algorithms require access to the full Hessian matrix. A recent line of work [Carmon et al., 2016, Agarwal et al., 2017, Carmon and Duchi, 2016] shows that the requirement of full Hessian access can be relaxed to Hessian-vector products, which can be computed efficiently in many machine learning applications. For pure gradient-based algorithms without access to Hessian information, Ge et al. [2015] show that adding perturbation in each iteration suffices to escape saddle points in polynomial time. When smoothness parameters are all dimension independent, Levy [2016] analyzed a normalized form of gradient descent with perturbation, and improved the dimension dependence to $O(d^3)$. This dependence has been further improved in recent work [Jin et al., 2017] to polylog$(d)$ via perturbed gradient descent (Algorithm 1).

## 1.2 Organization

This paper is organized as follows. In Section 2, we introduce the formal problem setting and background. In Section 3, we discuss some pathological examples and "un-natural" initialization schemes under which the gradient descent fails to escape strict saddle points in polynomial time. In Section 4, we show that even under a fairly natural initialization scheme, gradient descent still needs exponential time to escape all saddle points whereas perturbed gradient descent is able to do so in polynomial time. We provide empirical illustrations in Section 5 and conclude in Section 6. We place most of our detailed proofs in the Appendix.

## 2 Preliminaries

Let $\|\cdot\|_2$ denote the Euclidean norm of a finite-dimensional vector in $\mathbb{R}^d$. For a symmetric matrix $A$, let $\|A\|_{op}$ denote its operator norm and $\lambda_{\min}(A)$ its smallest eigenvalue. For a function $f : \mathbb{R}^d \to \mathbb{R}$, let $\nabla f(\cdot)$ and $\nabla^2 f(\cdot)$ denote its gradient vector and Hessian matrix. Let $\mathbb{B}_x(r)$ denote the $d$-dimensional $\ell_2$ ball centered at $x$ with radius $r$, $[-1,1]^d$ denote the $d$-dimensional cube centered at $0$ with side-length 2, and $B_\infty(x, R) = x + [-R, R]^d$ denote the $d$-dimensional cube centered at $x$ with side-length $2R$. We also use $O(\cdot)$, and $\Omega(\cdot)$ as standard Big-O and Big-Omega notation, only hiding absolute constants.

Throughout the paper we consider functions that satisfy the following smoothness assumptions.

**Definition 2.1.** *A function $f(\cdot)$ is B-bounded if for any* $\mathbf{x} \in \mathbb{R}^d$:

$$|f(\mathbf{x})| \leq B.$$

**Definition 2.2.** *A differentiable function $f(\cdot)$ is $\ell$-gradient Lipschitz if for any* $\mathbf{x}, \mathbf{y} \in \mathbb{R}^d$:

$$\|\nabla f(\mathbf{x}) - \nabla f(\mathbf{y})\|_2 \leq \ell \|\mathbf{x} - \mathbf{y}\|_2.$$

**Definition 2.3.** *A twice-differentiable function $f(\cdot)$ is $\rho$-Hessian Lipschitz if for any* $\mathbf{x}, \mathbf{y} \in \mathbb{R}^d$:

$$\left\|\nabla^2 f(\mathbf{x}) - \nabla^2 f(\mathbf{y})\right\|_{op} \leq \rho \|\mathbf{x} - \mathbf{y}\|_2.$$

Intuitively, definition 2.1 says function value is both upper and lower bounded; definition 2.2 and 2.3 state the gradients and Hessians of function can not change dramatically if two points are close by. Definition 2.2 is a standard asssumption in the optimization literature, and definition 2.3 is also commonly assumed when studying saddle points and local minima.

Our goal is to escape saddle points. The saddle points discussed in this paper are assumed to be "strict" [Ge et al., 2015]:

**Definition 2.4.** *A saddle point $\mathbf{x}^*$ is called an $\alpha$-strict saddle point if there exists some $\alpha > 0$ such that* $\|\nabla f(\mathbf{x}^*)\|_2 = 0$ and $\lambda_{min}\left(\nabla^2 f(\mathbf{x}^*)\right) \leq -\alpha$.

That is, a strict saddle point must have an escaping direction so that the eigenvalue of the Hessian along that direction is strictly negative. It turns out that for many non-convex problems studied in machine learning, all saddle points are strict (see Section 1 for more details).

To escape strict saddle points and converge to local minima, we can equivalently study the approximation of second-order stationary points. For $\rho$-Hessian Lipschitz functions, such points are defined as follows by Nesterov and Polyak [2006]:

---

**Algorithm 1** Perturbed Gradient Descent [Jin et al., 2017]

---

1: **Input:** $\mathbf{x}^{(0)}$, step size $\eta$, perturbation radius $r$, time interval $t_{\text{thres}}$, gradient threshold $g_{\text{thres}}$.
2: $t_{\text{noise}} \leftarrow -t_{\text{thres}} - 1$.
3: **for** $t = 1, 2, \cdots$ **do**
4:    **if** $\|\nabla f(x^t)\|_2 \leq g_{\text{thres}}$ and $t - t_{\text{noise}} > t_{\text{thres}}$ **then**
5:       $\mathbf{x}^{(t)} \leftarrow \mathbf{x}^{(t)} + \xi^t$, $\xi^t \sim \text{unif}(\mathbb{B}_0(r))$, $t_{\text{noise}} \leftarrow t$,
6:    **end if**
7:    $\mathbf{x}^{(t+1)} \leftarrow \mathbf{x}^{(t)} - \eta \nabla f(\mathbf{x}^{(t)})$.
8: **end for**

---

**Definition 2.5.** *A point $\mathbf{x}$ is a called a second-order stationary point if $\|\nabla f(\mathbf{x})\|_2 = 0$ and $\lambda_{min}(\nabla^2 f(\mathbf{x})) \geq 0$. We also define its $\epsilon$-version, that is, an $\epsilon$-second-order stationary point for some $\epsilon > 0$, if point $\mathbf{x}$ satisfies $\|\nabla f(\mathbf{x})\|_2 \leq \epsilon$ and $\lambda_{min}(\nabla^2 f(\mathbf{x})) \geq -\sqrt{\rho\epsilon}$.*

Second-order stationary points must have a positive semi-definite Hessian in additional to a vanishing gradient. Note if all saddle points $\mathbf{x}^*$ are strict, then second-order stationary points are exactly equivalent to local minima.

In this paper, we compare gradient descent and one of its variants—the perturbed gradient descent algorithm (Algorithm 1) proposed by Jin et al. [2017]. We focus on the case where the step size satisfies $\eta < 1/\ell$, which is commonly required for finding a minimum even in the convex setting [Nesterov, 2013].

The following theorem shows that if GD with random initialization converges, then it will converge to a second-order stationary point almost surely.

**Theorem 2.6** ([Lee et al., 2016] ). *Suppose that $f$ is $\ell$-gradient Lipschitz, has continuous Hessian, and step size $\eta < \frac{1}{\ell}$. Furthermore, assume that gradient descent converges, meaning $\lim_{t\to\infty} \mathbf{x}^{(t)}$ exists, and the initialization distribution $\nu$ is absolutely continuous with respect to Lebesgue measure. Then $\lim_{t\to\infty} \mathbf{x}^{(t)} = \mathbf{x}^*$ with probability one, where $\mathbf{x}^*$ is a second-order stationary point.*

The assumption that gradient descent converges holds for many non-convex functions (including all the examples considered in this paper). This assumption is used to avoid the case when $\|\mathbf{x}^{(t)}\|_2$ goes to infinity, so $\lim_{t\to\infty} \mathbf{x}^{(t)}$ is undefined.

Note the Theorem 2.6 only provides limiting behavior without specifying the convergence rate. On the other hand, if we are willing to add perturbations, the following theorem not only establishes convergence but also provides a sharp convergence rate:

**Theorem 2.7** ([Jin et al., 2017]). *Suppose $f$ is $B$-bounded, $\ell$-gradient Lipschitz, $\rho$-Hessian Lipschitz. For any $\delta > 0$, $\epsilon \leq \frac{\ell^2}{\rho}$, there exists a proper choice of $\eta, r, t_{\text{thres}}, g_{\text{thres}}$ (depending on $B, \ell, \rho, \delta, \epsilon$) such that Algorithm 1 will find an $\epsilon$-second-order stationary point, with at least probability $1 - \delta$, in the following number of iterations:*

$$O\left(\frac{\ell B}{\epsilon^2} \log^4\left(\frac{d\ell B}{\epsilon^2 \delta}\right)\right).$$

This theorem states that with proper choice of hyperparameters, perturbed gradient descent can consistently escape strict saddle points and converge to second-order stationary point in a polynomial number of iterations.

## 3 Warmup: Examples with "Un-natural" Initialization

The convergence result of Theorem 2.6 raises the following question: can gradient descent find a second-order stationary point in a polynomial number of iterations? In this section, we discuss two very simple and intuitive counter-examples for which gradient descent with random initialization requires an exponential number of steps to escape strict saddle points. We will also explain that, however, these examples are unnatural and pathological in certain ways, thus unlikely to arise in practice. A more sophisticated counter-example with natural initialization and non-pathological behavior will be given in Section 4.

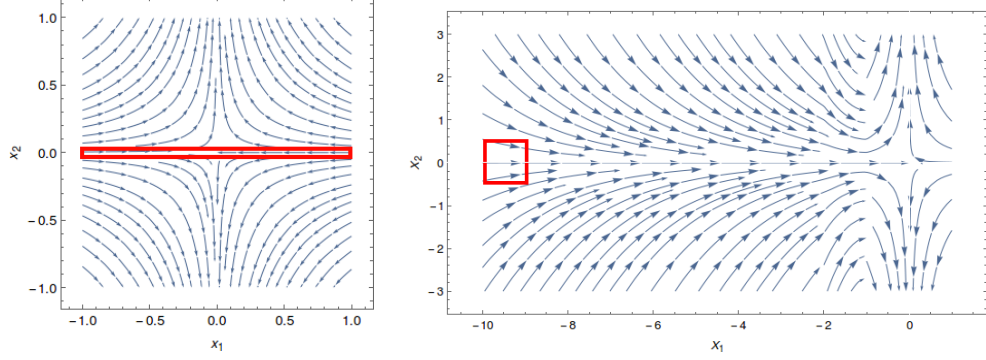

(a) Negative Gradient Field of $f(\mathbf{x}) = x_1^2 - x_2^2$.

(b) Negative Gradient Field for function defined in Equation (2).

Figure 1: If the initialization point is in red rectangle then it takes GD a long time to escape the neighborhood of saddle point $(0,0)$.

**Initialize uniformly within an extremely thin band.** Consider a two-dimensional function $f$ with a strict saddle point at $(0,0)$. Suppose that inside the neighborhood $U = [-1,1]^2$ of the saddle point, function is locally quadratic $f(x_1, x_2) = x_1^2 - x_2^2$, For GD with $\eta = \frac{1}{4}$, the update equation can be written as

$$x_1^{(t+1)} = \frac{x_1^{(t)}}{2} \quad \text{and} \quad x_2^{(t+1)} = \frac{3x_2^{(t)}}{2}.$$

If we initialize uniformly within $[-1,1] \times [-(\frac{3}{2})^{-\exp(\frac{1}{\epsilon})}, (\frac{3}{2})^{-\exp(\frac{1}{\epsilon})}]$ then GD requires at least $\exp(\frac{1}{\epsilon})$ steps to get out of neighborhood $U$, and thereby escape the saddle point. See Figure 1a for illustration. Note that in this case the initialization region is *exponentially* thin (only of width $2 \cdot (\frac{3}{2})^{-\exp(\frac{1}{\epsilon})}$). We would seldom use such an initialization scheme in practice.

**Initialize far away.** Consider again a two-dimensional function with a strict saddle point at $(0,0)$. This time, instead of initializing in a extremely thin band, we construct a very long slope so that a relatively large initialization region necessarily converges to this extremely thin band. Specifically, consider a function in the domain $[-\infty, 1] \times [-1, 1]$ that is defined as follows:

$$f(x_1, x_2) = \begin{cases} x_1^2 - x_2^2 & \text{if} \quad -1 < x_1 < 1 \\ -4x_1 + x_2^2 & \text{if} \quad x_1 < -2 \\ h(x_1, x_2) & \text{otherwise,} \end{cases} \tag{2}$$

where $h(x_1, x_2)$ is a smooth function connecting region $[-\infty, -2] \times [-1, 1]$ and $[-1, 1] \times [-1, 1]$ while making $f$ have continuous second derivatives and ensuring $x_2$ does not suddenly increase when $x_1 \in [-2, -1]$.[2] For GD with $\eta = \frac{1}{4}$, when $-1 < x_1 < 1$, the dynamics are

$$x_1^{(t+1)} = \frac{x_1^{(t)}}{2} \quad \text{and} \quad x_2^{(t+1)} = \frac{3x_2^{(t)}}{2},$$

and when $x_1 < -2$ the dynamics are

$$x_1^{(t+1)} = x_1^{(t)} + 1 \quad \text{and} \quad x_2^{(t+1)} = \frac{x_2^{(t)}}{2}.$$

Suppose we initialize uniformly within $[-R-1, -R+1] \times [-1, 1]$, for $R$ large. See Figure 1b for an illustration. Letting $t$ denote the first time that $x_1^{(t)} \geq -1$, then approximately we have $t \approx R$ and so $x_2^{(t)} \approx x_2^{(0)} \cdot (\frac{1}{2})^R$. From the previous example, we know that if $(\frac{1}{2})^R \approx (\frac{3}{2})^{-\exp\frac{1}{\epsilon}}$, that is $R \approx \exp\frac{1}{\epsilon}$, then GD will need exponential time to escape from the neighborhood $U = [-1, 1] \times [-1, 1]$ of the saddle point $(0, 0)$. In this case, we require an initialization region leading to a saddle point at distance $R$ which is exponentially large. In practice, it is unlikely that we would initialize exponentially far away from the saddle points or optima.

# 4 Main Result

In the previous section we have shown that gradient descent takes exponential time to escape saddle points under "un-natural" initialization schemes. Is it possible for the same statement to hold even under "natural" initialization schemes and non-pathological functions? The following theorem confirms this:

**Theorem 4.1** (Uniform initialization over a unit cube). *Suppose the initialization point is uniformly sampled from $[-1, 1]^d$. There exists a function $f$ defined on $\mathbb{R}^d$ that is $B$-bounded, $\ell$-gradient Lipschitz and $\rho$-Hessian Lipschitz with parameters $B, \ell, \rho$ at most $\mathrm{poly}(d)$ such that:*

1. *with probability one, gradient descent with step size $\eta \leq 1/\ell$ will be $\Omega(1)$ distance away from any local minima for any $T \leq e^{\Omega(d)}$.*

2. *for any $\epsilon > 0$, with probability $1 - e^{-d}$, perturbed gradient descent (Algorithm 1) will find a point $x$ such that $\|x - x^*\|_2 \leq \epsilon$ for some local minimum $x^*$ in $\mathrm{poly}(d, \frac{1}{\epsilon})$ iterations.*

**Remark:** As will be apparent in the next section, in the example we constructed, there are $2^d$ symmetric local minima at locations $(\pm c \ldots, \pm c)$, where $c$ is some constant. The saddle points are of the form $(\pm c, \ldots, \pm c, 0, \ldots, 0)$. Both algorithms will travel across $d$ neighborhoods of saddle points before reaching a local minimum. For GD, the number of iterations to escape the $i$-th saddle point increases as $\kappa^i$ ($\kappa$ is a multiplicative factor larger than 1), and thus GD requires exponential time to escape $d$ saddle points. On the other hand, PGD takes about the same number of iterations to escape each saddle point, and so escapes the $d$ saddle points in polynomial time. Notice that $B, \ell, \rho = O(\mathrm{poly}(d))$, so this does not contradict Theorem 2.7.

We also note that in our construction, the local minimizers are outside the initialization region. We note this is common especially for unconstrained optimization problems, where the initialization is usually uniform on a rectangle or isotropic Gaussian. Due to isoperimetry, the initialization concentrates in a thin shell, but frequently the final point obtained by the optimization algorithm is not in this shell.

It turns out in our construction, the only second-order stationary points in the path are the final local minima. Therefore, we can also strengthen Theorem 4.1 to provide a negative result for approximating $\epsilon$-second-order stationary points as well.

**Corollary 4.2.** *Under the same initialization as in Theorem 4.1, there exists a function $f$ satisfying the requirements of Theorem 4.1 such that for some $\epsilon = 1/\mathrm{poly}(d)$, with probability one, gradient descent with step size $\eta \leq 1/\ell$ will not visit any $\epsilon$-second-order stationary point in $T \leq e^{\Omega(d)}$.*

The corresponding positive result that PGD to find $\epsilon$-second-order stationary point in polynomial time immediately follows from Theorem 2.7.

The next result shows that gradient descent does not fail due to the special choice of initializing uniformly in $[-1, 1]^d$. For a large class of initialization distributions $\nu$, we can generalize Theorem 4.1 to show that gradient descent with random initialization $\nu$ requires exponential time, and perturbed gradient only requires polynomial time.

**Corollary 4.3.** *Let $B_\infty(\mathbf{z}, R) = \{\mathbf{z}\} + [-R, R]^d$ be the $\ell_\infty$ ball of radius $R$ centered at $\mathbf{z}$. Then for any initialization distribution $\nu$ that satisfies $\nu(B_\infty(\mathbf{z}, R)) \geq 1 - \delta$ for any $\delta > 0$, the conclusion of Theorem 4.1 holds with probability at least $1 - \delta$.*

That is, as long as most of the mass of the initialization distribution $\nu$ lies in some $\ell_\infty$ ball, a similar conclusion to that of Theorem 4.1 holds with high probability. This result applies to random Gaussian initialization, $\nu = \mathcal{N}(0, \sigma^2 \mathbf{I})$, with mean 0 and covariance $\sigma^2 \mathbf{I}$, where $\nu(B_\infty(0, \sigma \log d)) \geq 1 - 1/\mathrm{poly}(d)$.

## 4.1 Proof Sketch

In this section we present a sketch of the proof of Theorem 4.1. The full proof is presented in the Appendix. Since the polynomial-time guarantee for PGD is straightforward to derive from Jin et al. [2017], we focus on showing that GD needs an exponential number of steps. We rely on the following key observation.

**Key observation: escaping two saddle points sequentially.** Consider, for $L > \gamma > 0$,

$$f(x_1, x_2) = \begin{cases} -\gamma x_1^2 + L x_2^2 & \text{if } x_1 \in [0,1], x_2 \in [0,1] \\ L(x_1 - 2)^2 - \gamma x_2^2 & \text{if } x_1 \in [1,3], x_2 \in [0,1] \\ L(x_1 - 2)^2 + L(x_2 - 2)^2 & \text{if } x_1 \in [1,3], x_2 \in [1,3] \end{cases} \tag{3}$$

Note that this function is not continuous. In the next paragraph we will modify it to make it smooth and satisfy the assumptions of the Theorem but useful intuition is obtained using this discontinuous function. The function has an optimum at $(2,2)$ and saddle points at $(0,0)$ and $(2,0)$. We call $[0,1] \times [0,1]$ the neighborhood of $(0,0)$ and $[1,3] \times [0,1]$ the neighborhood of $(2,0)$. Suppose the initialization $(x^{(0)}, y^{(0)})$ lies in $[0,1] \times [0,1]$. Define $t_1 = \min_{x_1^{(t)} \geq 1} t$ to be the time of first departure from the neighborhood of $(0,0)$ (thereby escaping the first saddle point). By the dynamics of gradient descent, we have

$$x_1^{(t_1)} = (1 + 2\eta\gamma)^{t_1} x_1^{(0)}, \quad x_2^{(t_1)} = (1 - 2\eta L)^{t_1} x_2^{(0)}.$$

Next we calculate the number of iterations such that $x_2 \geq 1$ and the algorithm thus leaves the neighborhood of the saddle point $(2,0)$ (thus escaping the second saddle point). Letting $t_2 = \min_{x_2^{(t)} \geq 1} t$, we have:

$$x_2^{(t_1)}(1 + 2\eta\gamma)^{t_2 - t_1} = (1 + 2\eta\gamma)^{t_2 - t_1}(1 - 2\eta L)^{t_1} x_2^{(0)} \geq 1.$$

We can lower bound $t_2$ by

$$t_2 \geq \frac{2\eta(L + \gamma)t_1 + \log(\frac{1}{x_2^0})}{2\eta\gamma} \geq \frac{L + \gamma}{\gamma} t_1.$$

The key observation is that the number of steps to escape the second saddle point is $\frac{L+\gamma}{\gamma}$ times the number of steps to escape the first one.

**Spline: connecting quadratic regions.** To make our function smooth, we create buffer regions and use splines to interpolate the discontinuous parts of Equation (3). Formally, we consider the following function, for some fixed constant $\tau > 1$:

$$f(x_1, x_2) = \begin{cases} -\gamma x_1^2 + L x_2^2 & \text{if } x_1 \in [0, \tau], x_2 \in [0, \tau] \\ g(x_1, x_2) & \text{if } x_1 \in [\tau, 2\tau], x_2 \in [0, \tau] \\ L(x_1 - 4\tau)^2 - \gamma x_2^2 - \nu & \text{if } x_1 \in [2\tau, 6\tau], x_2 \in [0, \tau] \\ L(x_1 - 4\tau)^2 + g_1(x_2) - \nu & \text{if } x_1 \in [2\tau, 6\tau], x_2 \in [\tau, 2\tau] \\ L(x_1 - 4\tau)^2 + L(x_2 - 4\tau)^2 - 2\nu & \text{if } x_1 \in [2\tau, 6\tau], x_2 \in [2\tau, 6\tau], \end{cases} \tag{4}$$

where $g, g_1$ are spline polynomials and $\nu > 0$ is a constant defined in Lemma B.2. In this case, there are saddle points at $(0,0)$, and $(4\tau, 0)$ and the optimum is at $(4\tau, 4\tau)$. Intuitively, $[\tau, 2\tau] \times [0, \tau]$ and $[2\tau, 6\tau] \times [\tau, 2\tau]$ are buffer regions where we use splines ($g$ and $g_1$) to transition between regimes and make $f$ a smooth function. Also in this region there is no stationary point and the smoothness assumptions are still satisfied in the theorem. Figure. 2a shows the surface and stationary points of this function. We call the union of the regions defined in Equation (4) a *tube*.

**From two saddle points to $d$ saddle points.** We can readily adapt our construction of the tube to $d$ dimensions, such that the function is smooth, the location of saddle points are $(0, \ldots, 0)$, $(4\tau, 0, \ldots, 0)$, ..., $(4\tau, \ldots, 4\tau, 0)$, and optimum is at $(4\tau, \ldots, 4\tau)$. Let $t_i$ be the number of step to escape the neighborhood of the $i$-th saddle point. We generalize our key observation to this case and obtain $t_{i+1} \geq \frac{L+\gamma}{\gamma} \cdot t_i$ for all $i$. This gives $t_d \geq (\frac{L+\gamma}{\gamma})^d$ which is exponential time. Figure 2b shows the tube and trajectory of GD.

**Mirroring trick: from tube to octopus.** In the construction thus far, the saddle points are all on the boundary of tube. To avoid the difficulties of constrained non-convex optimization, we would like to make all saddle points be interior points of the domain. We use a simple mirroring trick; i.e., for every coordinate $x_i$ we reflect $f$ along its axis. See Figure 2c for an illustration in the case $d = 2$.

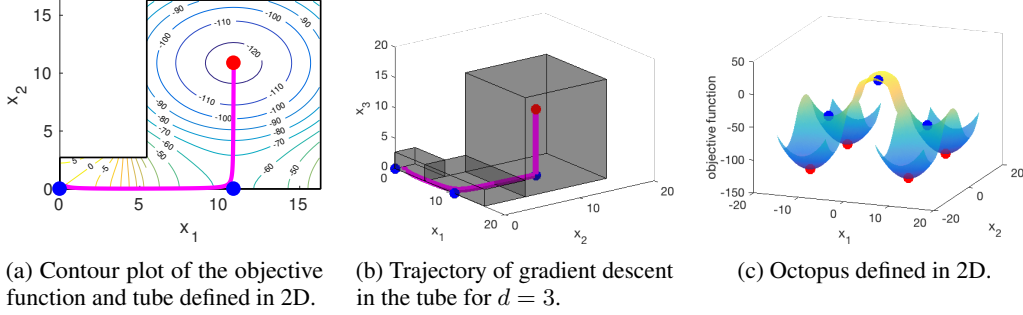

(a) Contour plot of the objective function and tube defined in 2D.

(b) Trajectory of gradient descent in the tube for $d = 3$.

(c) Octopus defined in 2D.

Figure 2: Graphical illustrations of our counter-example with $\tau = e$. The blue points are saddle points and the red point is the minimum. The pink line is the trajectory of gradient descent.

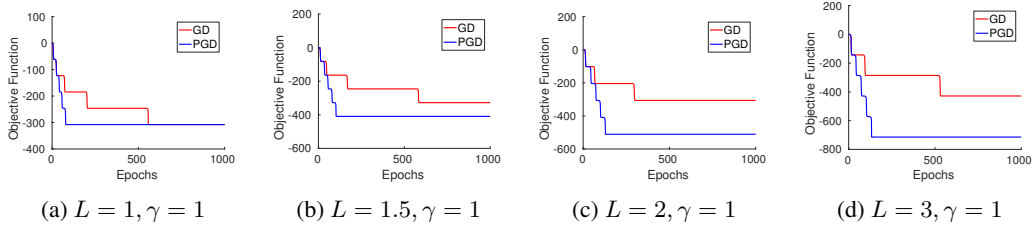

(a) $L = 1, \gamma = 1$     (b) $L = 1.5, \gamma = 1$     (c) $L = 2, \gamma = 1$     (d) $L = 3, \gamma = 1$

Figure 3: Performance of GD and PGD on our counter-example with $d = 5$.

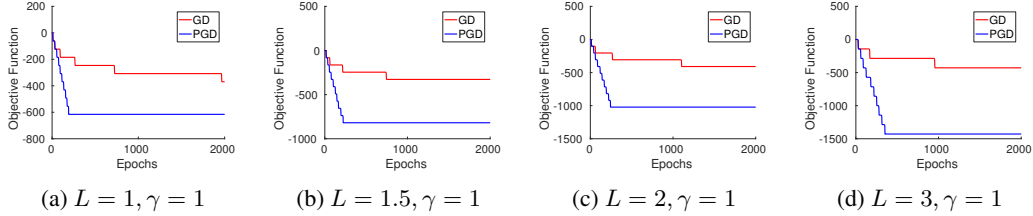

(a) $L = 1, \gamma = 1$     (b) $L = 1.5, \gamma = 1$     (c) $L = 2, \gamma = 1$     (d) $L = 3, \gamma = 1$

Figure 4: Performance of GD and PGD on our counter-example with $d = 10$

**Extension: from octopus to $\mathbb{R}^d$.** Up to now we have constructed a function defined on a closed subset of $\mathbb{R}^d$. The last step is to extend this function to the entire Euclidean space. Here we apply the classical Whitney Extension Theorem (Theorem B.3) to finish our construction. We remark that the Whitney extension may lead to more stationary points. However, we will demonstrate in the proof that GD and PGD stay within the interior of "octopus" defined above, and hence cannot converge to any other stationary point.

## 5 Experiments

In this section we use simulations to verify our theoretical findings. The objective function is defined in (14) and (15) in the Appendix. In Figures 3 and Figure 4, GD stands for gradient descent and PGD stands for Algorithm 1. For both GD and PGD we let the stepsize $\eta = \frac{1}{4L}$. For PGD, we choose $t_{\text{thres}} = 1$, $g_{\text{thres}} = \frac{\gamma e}{100}$ and $r = \frac{e}{100}$. In Figure 3 we fix dimension $d = 5$ and vary $L$ as considered in Section 4.1; similarly in Figure 4 we choose $d = 10$ and vary $L$. First notice that in all experiments, PGD converges faster than GD as suggested by our theorems. Second, observe the "horizontal" segment in each plot represents the number of iterations to escape a saddle point. For GD the length of the segment grows at a fixed rate, which coincides with the result mentioned at the beginning for Section 4.1 (that the number of iterations to escape a saddle point increase at each time with a multiplicative factor $\frac{L+\gamma}{\gamma}$). This phenomenon is also verified in the figures by the fact that as the ratio $\frac{L+\gamma}{\gamma}$ becomes larger, the rate of growth of the number of iterations to escape increases. On the other hand, the number of iterations for PGD to escape is approximately constant ($\sim \frac{1}{\eta\gamma}$).

# 6    Conclusion

In this paper we established the failure of gradient descent to efficiently escape saddle points for general non-convex smooth functions. We showed that even under a very natural initialization scheme, gradient descent can require exponential time to converge to a local minimum whereas perturbed gradient descent converges in polynomial time. Our results demonstrate the necessity of adding perturbations for efficient non-convex optimization.

We expect that our results and constructions will naturally extend to a stochastic setting. In particular, we expect that with random initialization, general stochastic gradient descent will need exponential time to escape saddle points in the worst case. However, if we add perturbations per iteration or the inherent randomness is non-degenerate in every direction (so the covariance of noise is lower bounded), then polynomial time is known to suffice [Ge et al., 2015].

One open problem is whether GD is inherently slow if the local optimum is inside the initialization region in contrast to the assumptions of initialization we used in Theorem 4.1 and Corollary 4.3. We believe that a similar construction in which GD goes through the neighborhoods of $d$ saddle points will likely still apply, but more work is needed. Another interesting direction is to use our counter-example as a building block to prove a computational lower bound under an oracle model [Nesterov, 2013, Woodworth and Srebro, 2016].

This paper does not rule out the possibility for gradient descent to perform well for some non-convex functions with special structures. Indeed, for the matrix square-root problem, Jain et al. [2017] show that with reasonable random initialization, gradient updates will stay away from all saddle points, and thus converge to a local minimum efficiently. It is an interesting future direction to identify other classes of non-convex functions that gradient descent can optimize efficiently and not suffer from the negative results described in this paper.

# 7    Acknowledgements

S.S.D. and B.P. were supported by NSF grant IIS1563887 and ARPA-E Terra program. C.J. and M.I.J. were supported by the Mathematical Data Science program of the Office of Naval Research under grant number N00014-15-1-2670. J.D.L. was supported by ARO W911NF-17-1-0304. A.S. was supported by DARPA grant D17AP00001, AFRL grant FA8750-17-2-0212 and a CMU ProS-EED/BrainHub Seed Grant. The authors thank Rong Ge, Qing Qu, John Wright, Elad Hazan, Sham Kakade, Benjamin Recht, Nathan Srebro, and Lin Xiao for useful discussions. The authors thank Stephen Wright and Michael O'Neill for pointing out calculation errors in the older version.

## Footnotes

[1]Assuming that the smoothness parameters (see Definition 2.1- 2.3) are all independent of dimension.

[2] We can construct such a function using splines. See Appendix B.

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
