[Supplementary Material · gd_lower_bound_supp.pdf]

# A Proofs for Results in Section 4

In this section, we provide proofs for Theorem 4.1 and Corollary 4.3. The proof for Corollary 4.2 easily follows from the same construction as in Theorem 4.1, so we omit it here. For Theorem 4.1, we will prove each claim individually.

## A.1 Proof for Claim 1 of Theorem 4.1

**Outline of the proof.**   Our construction of the function is based on the intuition in Section 4.1. Note the function $f$ defined in (3) is 1) not continuous whereas we need a $C^2$ continuous function and 2) only defined on a subset of Euclidean space whereas we need a function defined on $\mathbb{R}^d$. To connect these quadratic functions, we use high-order polynomials based on spline theory. We connect $d$ such quadratic functions and show that GD needs exponential time to converge if $\mathbf{x}^{(0)} \in [0, 1]^d$. Next, to make all saddle points as interior point, we exploit symmetry and use a mirroring trick to create $2^d$ copies of the spline. This ensures that as long as the initialization is in $[-1, 1]^d$, gradient descent requires exponential steps. Lastly, we use the classical Whitney extension theorem [Whitney, 1934] to extend our function from a closed subset to $\mathbb{R}^d$.

**Step 1: The tube.**   We fix four constants $L = e$, $\gamma = 1$, $\tau = e$ and $\nu = -g_1(2\tau) + 4L\tau^2$ where $g_1$ is defined in Lemma B.2. We first construct a function $f$ and a closed subset $D_0 \subset \mathbb{R}^d$ such that if $\mathbf{x}^{(0)}$ is initialized in $[0, 1]^d$ then the gradient descent dynamics will get stuck around some saddle point for exponential time. Define the domain as:

$$D_0 = \bigcup_{i=1}^{d+1} \{x \in \mathbb{R}^d : 6\tau \geq x_1, \ldots x_{i-1} \geq 2\tau, 2\tau \geq x_i \geq 0, \tau \geq x_{i+1} \ldots, x_d \geq 0\}, \quad (5)$$

which $i = 1$ means $0 \leq x_1 \leq 2\tau$ and other coordinates are smaller than $\tau$, and $i = d + 1$ means that all coordinates are larger than $2\tau$. See Figure 5a for an illustration. Next we define the objective function as follows. For a given $i = 1, \ldots, d-1$, if $6\tau \geq x_1, \ldots x_{i-1} \geq 2\tau, \tau \geq x_i \geq 0, \tau \geq x_{i+1} \ldots, x_d \geq 0$, we have

$$f(\mathbf{x}) = \sum_{j=1}^{i-1} L(x_j - 4\tau)^2 - \gamma x_i^2 + \sum_{j=i+1}^{d} L x_j^2 - (i-1)\nu \triangleq f_{i,1}(\mathbf{x}), \quad (6)$$

and if $6\tau \geq x_1, \ldots x_{i-1} \geq 2\tau, 2\tau \geq x_i \geq \tau, \tau \geq x_{i+1} \ldots, x_d \geq 0$, we have

$$f(\mathbf{x}) = \sum_{j=1}^{i-1} L(x_j - 4\tau)^2 + g(x_i, x_{i+1}) + \sum_{j=i+2}^{d} L x_j^2 - (i-1)\nu \triangleq f_{i,2}(\mathbf{x}), \quad (7)$$

where the constant $\nu$ and the bivariate function $g$ are specified in Lemma B.2 to ensure $f$ is a $C^2$ function and satisfies the smoothness assumptions in Theorem 4.1. For $i = d$, we define the objective function as

$$f(\mathbf{x}) = \sum_{j=1}^{d-1} L(x_j - 4\tau)^2 - \gamma x_d^2 - (d-1)\nu \triangleq f_{d,1}(\mathbf{x}), \quad (8)$$

if $6\tau \geq x_1, \ldots x_{d-1} \geq 2\tau$ and $\tau \geq x_d \geq 0$ and

$$f(\mathbf{x}) = \sum_{j=1}^{d-1} L(x_j - 4\tau)^2 + g_1(x_d) - (d-1)\nu \triangleq f_{d,2}(\mathbf{x}) \quad (9)$$

if $6\tau \geq x_1, \ldots x_{d-1} \geq 2\tau$ and $2\tau \geq x_d \geq \tau$ where $g_1$ is defined in Lemma B.2. Lastly, if $6\tau \geq x_1, \ldots x_d \geq 2\tau$, we define

$$f(\mathbf{x}) = \sum_{j=1}^{d} L(x_j - 4\tau)^2 - d\nu \triangleq f_{d+1,1}(\mathbf{x}). \quad (10)$$

Figure. 5a shows an intersection surface (a slice along the $x_i$-$x_{i+1}$ plane) of this construction.

(a) The intersection surface of the Tube defined in Equation (5) (6)and (7) for $2\tau \le x_1, \ldots, x_{i-1} \le 6\tau, 0 \le x_{i+2} \le \tau$.

(b) The "octopus"-like domain we defined in Equation (12) and (13) for $d = 2$.

Figure 5: Illustration of intersection surfaces used in our construction.

**Remark A.1.** *As will be apparent in Theorem B.2, $g$ and $g_1$ are polynomials with degrees bounded by five, which implies that for $\tau \le x_i \le 2\tau$ and $0 \le x_{i+1} \le \tau$ the function values and derivatives of $g(x_i, x_{i+1})$ and $g(x_i)$ are bounded by $\mathrm{poly}(L)$; in particular, $\rho = \mathrm{poly}(L)$.*

**Remark A.2.** *In Theorem B.2 we show that the norms of the gradients of $g$ and $g_1$ gradients are strictly larger than zero by a constant ($\ge \gamma\tau$), which implies that for $\epsilon < \gamma\tau$, there is no $\epsilon$-second-order stationary point in the connection region. Further note that in the domain of the function defined in Eq. (6) and (8), the smallest eigenvalue of Hessian is $-2\gamma$. Therefore we know that if $\mathbf{x} \in D_0$ and $x_d \le 2\tau$, then $x$ cannot be an $\epsilon$-second-order stationary point for $\epsilon \le \frac{4\gamma^2}{\rho}$*

Now let us study the stationary points of this function. Technically, the differential is only defined on the interior of $D_0$. However in Steps 2 and 3, we provide a $C^2$ extension of $f$ to all of $\mathbb{R}^d$, so the lemma below should be interpreted as characterizing the critical points of this extended function $f$ in $D_0$. Using the analytic form of Eq. (6)- (10) and Remark A.2, we can easily identify the stationary points of $f$.

**Lemma A.3.** *For $f : D_0 \to R$ defined in Eq. (6) to Eq. (10), there is only one local optimum:*

$$\mathbf{x}^* = (4\tau, \ldots, 4\tau)^\top,$$

*and $d$ saddle points:*

$$(0, \ldots, 0)^\top, (4\tau, 0, \ldots, 0)^\top, \ldots, (4\tau, \ldots, 4\tau, 0)^\top.$$

Next we analyze the convergence rate of gradient descent. The following lemma shows that it takes exponential time for GD to achieve $x_d \ge 2\tau$.

**Lemma A.4.** *Let $\tau \ge e$ and $x^{(0)} \in [-1, 1]^d \cap D_0$. GD with $\eta \le \frac{1}{2L}$ and any $T \le \left(\frac{L+\gamma}{\gamma}\right)^{d-1}$ satisfies $x_d^{(T)} \le 2\tau$.*

*Proof.* Define $T_0 = 0$ and for $k = 1, \ldots, d$, let $T_k = \min\{t | x_k^{(t)} \ge 2\tau\}$ be the first time the iterate escapes the neighborhood of the $k$-th saddle point. We also define $T_k^\tau$ as the number of iterations inside the region

$$\{x_1, \ldots, x_{k-1} \ge 2\tau, \tau \le x_k \le 2\tau, 0 \le x_{k+\ell}, \ldots, x_d \le \tau\}.$$

First we bound $T_k^\tau$. Lemma B.2 shows $\frac{\partial g(x_k, x_{k+1})}{\partial x_k} \le -2\gamma\tau$ so after every gradient descent step, $x_k$ is increased by at least $2\eta\gamma\tau$. Therefore we can upper bound $T_k^\tau$ by

$$T_k^\tau \le \frac{2\tau - \tau}{2\eta\gamma\tau} = \frac{1}{2\eta\gamma}.$$

Note this bound holds for all $k$.

Next, we lower bound $T_1$. By definition, $T_1$ is the smallest number such that $x_1^{(T_1)} \geq 2\tau$ and using the definition of $T_1^\tau$ we know $x_1^{(T_1 - T_1^\tau)} \geq \tau$. By the gradient update equation, for $t = 1 \ldots, T_1 - T_1^\tau$, we have $x_1^t = (1 + 2\eta\gamma)^t x_1^0$. Thus we have:

$$x_1^{(0)} (1 + 2\eta\gamma)^{T_1 - T_1^\tau} \geq \tau$$

$$\Rightarrow T_1 - T_1^\tau \geq \frac{1}{2\eta\gamma} \log\left(\frac{\tau}{x_1^{(0)}}\right).$$

Since $x_0^1 \leq 1$ and $\tau \geq e$, we know $\log(\frac{\tau}{x_1^0}) \geq 1$. Therefore $T_1 - T_1^\tau \geq \frac{1}{\eta\gamma} \geq T_1^\tau$.

Next we show iterates generated by GD stay in $D_0$. If $\mathbf{x}^{(t)}$ satisfies $6\tau \geq x_1, \ldots x_{k-1} \geq 2\tau, \tau \geq x_k \geq 0, \tau \geq x_{k+1} \ldots, x_d \geq 0$, then for $1 \leq j \leq k$,

$$x_j^{(t+1)} = (1 - \eta L) x_j^{(t)} - 4\eta L\tau \in [2\tau, 6\tau],$$

for $j = k$,

$$x_j^{(t+1)} = (1 + 2\eta\gamma) x_j^{(t)} \in [0, 2\tau],$$

and for $j \geq k + 1$

$$x_j^{(t+1)} = (1 - 2\eta L) x_j^{(t)} \in [0, \tau].$$

Similarly, if $x^{(t)}$ satisfies $6\tau \geq x_1, \ldots x_{k-1} \geq 2\tau, 2\tau \geq x_k \geq \tau, \tau \geq x_{k+1} \ldots, x_d \geq 0$, the above arguments still hold for $j \leq k - 1$ and $j \geq k + 2$. For $j = k$, note that

$$x_j^{(t+1)} = x_j^{(t)} - \eta \frac{\partial g(x_j, x_{j+1})}{\partial x_j}$$

$$\leq x_j^{(t)} + 2\eta\gamma\tau \leq 6\tau,$$

where in the first inequality we have used Lemma B.2. For $j = k + 1$, by the dynamics of gradient descent, at $(T_k - T_k^\tau)$-th iteration, $x_{k+1}^{(T_k - T_k^\tau)} = x_{k+1}^{(0)} (1 - 2\eta L)^{T_k - T_k^\tau}$. Note Lemma B.2 shows in the region

$$\{x_1, \ldots, x_{k-1} \geq 2\tau, \tau \leq x_k \leq 2\tau, 0 \leq x_{k+1}, \ldots, x_d \leq \tau\},$$

we have

$$\frac{\partial f(x)}{\partial x_{k+1}} \geq -2\gamma x_{k+1}.$$

Putting this together we have the following upper bounds for $t = T_k - T_k^\tau + 1, \ldots, T_k$:

$$x_{k+1}^{(t)} \leq x_{k+1}^0 (1 - 2\eta L)^{(T_k - T_k^\tau)} \cdot (1 + 2\eta\gamma)^{t - (T_k - T_k^\tau)} \leq \tau, \tag{11}$$

which implies $\mathbf{x}^{(t)}$ is in $D_0$.

Next, let us calculate the relation between $T_k$ and $T_{k+1}$. By our definition of $T_k$ and $T_k^\tau$, we have:

$$x_{k+1}^{(T_k)} \leq x_{k+1}^{(0)} (1 - 2\eta L)^{T_k - T_k^\tau} \cdot (1 + 2\eta\gamma)^{T_k^\tau}.$$

For $T_{k+1}$, with the same logic we used for lower bounding $T_1$, we have

$$x_{k+1}^{(T_{k+1} - T_{k+1}^\tau)} \geq \tau$$

$$\Rightarrow x_{k+1}^{(T_k)} (1 + 2\eta\gamma)^{T_{k+1} - T_{k+1}^\tau - T_k} \geq \tau$$

$$\Rightarrow x_{k+1}^{(0)} (1 - 2\eta L)^{T_k - T_k^\tau} \cdot (1 + 2\eta\gamma)^{T_k^\tau} \cdot (1 + 2\eta\gamma)^{T_{k+1} - T_{k+1}^\tau - T_k} \geq \tau.$$

Taking logarithms on both sides and then using $\log(1 - \theta) \leq -\theta, \log(1 + \theta) \leq \theta$ for $0 \leq \theta \leq 1$, and $\eta \leq \frac{1}{2L}$, we have

$$2\eta\gamma \left(T_{k+1} - T_{k+1}^\tau - (T_k - T_k^\tau)\right) \geq \log\left(\frac{\tau}{x_{k+1}^0}\right) + 2\eta L (T_k - T_k^\tau)$$

$$\Rightarrow T_{k+1} - T_{k+1}^\tau \geq \frac{L + \gamma}{\gamma} (T_k - T_k^\tau)$$

In last step, we used the initialization condition whereby $\log\left(\frac{\tau}{x_{k+1}^0}\right) \geq 1 \geq 0$. Since $T_1 - T_1^\tau \geq \frac{1}{2\eta\gamma}$, to enter the region $x_1, \ldots, x_d \geq 2\tau$ we need $T_d$ iterations, which is lower bounded by

$$T_d \geq \frac{1}{2\eta\gamma} \cdot \left(\frac{L+\gamma}{\gamma}\right)^{d-1} \geq \left(\frac{L+\gamma}{\gamma}\right)^{d-1}.$$

$\square$

**Step 2: From the tube to the octopus.** We have shown that if $x^0 \in [-1,1]^d \cap D_0$, then gradient descent needs exponential time to approximate a second order stationary point. To deal with initialization points in $[-1,1]^d - D_0$, we use a simple mirroring trick; i.e., for each coordinate $x_i$, we create a mirror domain of $D_0$ and a mirror function according to $i$-th axis and then take union of all resulting reflections. Therefore, we end up with an "octopus" which has $2^d$ copies of $D_0$ and $[-1,1]^d$ is a subset of this "octopus." Figure 5b shows the construction for $d = 2$.

The mirroring trick is used mainly to make saddle points be interior points of the region (octopus) and ensure that the positive result of PGD (claim 2) will hold.

We now formalize this mirroring trick. For $a = 0, \ldots, 2^d - 1$, let $a_2$ denote its binary representation. Denote $a_2(0)$ as the indices of $a_2$ with digit 0 and $a_2(1)$ as those that are 1. Now we define the domain

$$D_a = \bigcup_{i=1}^{d} \{x \in \mathbb{R}^d : x_i \geq 0 \text{ if } i \in a_2(0), x_i \leq 0 \text{ otherwise },$$
$$6\tau \geq |x_1| \ldots, |x_{i-1}| \geq 2\tau, |x_i| \leq 2\tau, |x_{i+1}| \ldots, |x_d| \leq \tau\}, \tag{12}$$

$$D = \bigcup_{a=0}^{2^d-1} D_a. \tag{13}$$

Note this is a closed subset of $\mathbb{R}^d$ and $[-1,1]^d \subset D$. Next we define the objective function. For $i = 1, \ldots, d-1$, if $6\tau \geq |x_1|, \ldots, |x_{i-1}| \geq 2\tau, |x_i| \leq \tau, |x_{i+1}| \ldots, |x_d| \leq \tau$:

$$f(\mathbf{x}) = \sum_{j \leq i-1, j \in a_2(0)} L(x_j - 4\tau)^2 + \sum_{j \leq i-1, j \in a_2(1)} L(x_j + 4\tau)^2 - \gamma x_i^2$$
$$+ \sum_{j=i+1}^{d} Lx_j^2 - (i-1)\nu, \tag{14}$$

and if $6\tau \geq |x_1|, \ldots, |x_{i-1}| \geq 2\tau, \tau \leq |x_i| \leq 2\tau, |x_{i+1}|, \ldots, |x_d| \leq \tau$:

$$f(\mathbf{x}) = \sum_{j \leq i-1, j \in a_2(0)} L(x_j - 4\tau)^2 + \sum_{j \leq i-1, j \in a_2(1)} L(x_j + 4\tau)^2 + G(x_i, x_{i+1})$$
$$+ \sum_{j=i+2}^{d} Lx_j^2 - (i-1)\nu, \tag{15}$$

where

$$G(x_i, x_{i+1}) = \begin{cases} g(x_i, x_{i+1}) \text{ if } i \in a_2(0) \\ g(-x_i, x_{i+1}) \text{ if } i \in a_2(1). \end{cases}$$

For $i = d$, if $6\tau \geq |x_1|, \ldots, |x_{i-1}| \geq 2\tau, |x_i| \leq \tau$:

$$f(\mathbf{x}) = \sum_{j \leq i-1, j \in a_2(0)} L(x_j - 4\tau)^2 + \sum_{j \leq i-1, j \in a_2(1)} L(x_j + 4\tau)^2 - \gamma x_i^2 - (i-1)\nu, \tag{16}$$

and if $6\tau \geq |x_1| \ldots, |x_{i-1}| \geq 2\tau, \tau \leq |x_i| \leq 2\tau$:

$$f(\mathbf{x}) = \sum_{j \leq i-1, j \in a_2(0)} L(x_j - 4\tau)^2 + \sum_{j \leq i-1, j \in a_2(1)} L(x_j + 4\tau)^2 + G_1(x_i) - (i-1)\nu, \tag{17}$$

where

$$G_1(x_i) = \begin{cases} g_1(x_i) \text{ if } i \in a_2(0) \\ g_1(-x_i) \text{ if } i \in a_2(1). \end{cases}$$

Lastly, if $6\tau \geq |x_1|, \ldots, |x_d| \geq 2\tau$:

$$f(\mathbf{x}) = \sum_{j \leq i-1, j \in a_2(0)} L(x_j - 4\tau)^2 + \sum_{j \leq i-1, j \in a_2(1)} L(x_j + 4\tau)^2 - d\nu. \qquad (18)$$

Note that if a coordinate $x_i$ satisfies $|x_i| \leq \tau$, the function defined in Eq. (14) to (17) is an even function (fix all $x_j$ for $j \neq i$, $f(\ldots, x_i, \ldots) = f(\ldots, -x_i, \ldots)$) so $f$ preserves the smoothness of $f_0$. By symmetry, mirroring the proof of Lemma A.4 for $D_a$ for $a = 1, \ldots, 2^d - 1$ we have the following lemma:

**Lemma A.5.** *Choosing $\tau = e$, if $\mathbf{x}^{(0)} \in [-1, 1]^d$ then for gradient descent with $\eta \leq \frac{1}{2L}$ and any $T \leq \left(\frac{L+\gamma}{\gamma}\right)^{d-1}$, we have $x_d^{(T)} \leq 2\tau$.*

**Step 3: From the octopus to $\mathbb{R}^d$.** It remains to extend $f$ from $D$ to $\mathbb{R}^d$. Here we use the classical Whitney extension theorem (Theorem B.3) to obtain our final function $F$. Applying Theorem B.3 to $f$ we have that there exists a function $F$ defined on $\mathbb{R}^d$ which agrees with $f$ on $D$ and the norms of its function values and derivatives of all orders are bounded by $O(\text{poly}(d))$. Note that this extension may introduce new stationary points. However, as we have shown previously, GD never leaves $D$ so we can safely ignore these new stationary points. We have now proved the negative result regarding gradient descent.

## A.2 Proof for Claim 2 of Theorem 4.1

To show that PGD approximates a local minimum in polynomial time, we first apply Theorem 2.7 which shows that PGD finds an $\epsilon$-second-order stationary point. Remark A.2 shows in $D$, every $\epsilon$-second-order stationary point is $\epsilon$ close to a local minimum. Thus, it suffices to show iterates of PGD stay in $D$. We will prove the following two facts: 1) after adding noise, $\mathbf{x}$ is still in $D$, and 2) until the next time we add noise, $\mathbf{x}$ is in $D$.

For the first fact, using the choices of $g_{\text{thres}}$ and $r$ in Jin et al. [2017] we can pick $\epsilon$ polynomially small enough so that $g_{\text{thres}} \leq \frac{\gamma\tau}{10}$ and $r \leq \frac{\tau}{20}$, which ensures there is no noise added when there exists a coordinate $x_i$ with $\tau \leq x_i \leq 2\tau$. Without loss of generality, suppose that in the region

$$\{x_1, \ldots, x_{k-1} \geq 2\tau, 0 \leq x_k, \ldots, x_d \leq \tau\},$$

we have $\|\nabla f(\mathbf{x})\|_2 \leq g_{\text{thres}} \leq \frac{\gamma\tau}{10}$, which implies $|x_j - 4\tau| \leq \frac{\tau}{20}$ for $j = 1, \ldots, k-1$, and $x_j \leq \frac{\tau}{20}$ for $j = k, \ldots, d$. Therefore, $\left|(\mathbf{x} + \xi)_j - 4\tau\right| \leq \frac{\tau}{10}$ for $j = 1, \ldots, k-1$ and

$$\left|(\mathbf{x} + \xi)_j\right| \leq \frac{\tau}{10} \qquad (19)$$

for $j = k, \ldots, d$.

For the second fact suppose at the $t'$-th iteration we add noise. Now without loss of generality, suppose that after adding noise, $\mathbf{x}^{(t')} \geq 0$, and by the first fact $\mathbf{x}^{t'}$ is in the region

$$\left\{x_1, \ldots, x_{i-1} \geq 2\tau, 0 \leq x_i \leq \ldots, x_d \leq \frac{\tau}{10}\right\}.$$

Now we use the same argument as for proving GD stays in $D$. Suppose at $t''$-th iteration we add noise again. Then for $t' < t < t''$, we have that if $\mathbf{x}^{(t)}$ satisfies $6\tau \geq x_1, \ldots x_{k-1} \geq 2\tau, \tau \geq x_k \geq 0, \tau \geq x_{k+1} \ldots, x_d \geq 0$, then for $1 \leq j \leq k$,

$$x_j^{(t+1)} = (1 - \eta L) x_j^{(t)} - 4\eta L\tau \in [2\tau, 6\tau],$$

for $j = k$,

$$x_j^{(t+1)} = (1 + 2\eta\gamma) x_j^{(t)} \in [0, 2\tau],$$

and for $j \geq k+1$

$$x_j^{(t+1)} = (1 - 2\eta L)\, x_j^{(t)} \in [0, \tau]\,.$$

Similarly, if $x^{(t)}$ satisfies $6\tau \geq x_1, \ldots x_{k-1} \geq 2\tau, 2\tau \geq x_k \geq \tau, \tau \geq x_{k+1} \ldots, x_d \geq 0$, the above arguments still hold for $j \leq k-1$ and $j \geq k+2$. For $j = k$, note that

$$x_j^{(t+1)} = x_j^{(t)} - \eta \frac{\partial g\,(x_j, x_{j+1})}{\partial x_j}$$

$$\leq x_j^{(t)} + 4\eta L \tau \leq 6\tau,$$

where the first inequality we have used Lemma B.2.

For $j = k+1$, by the dynamics of gradient descent, at the $(T_k - T_k^\tau)$-th iteration, $x_{k+1}^{(T_k - T_k^\tau)} = x_{k+1}^{(t')} (1 - 2\eta L)^{T_k - T_k^\tau - t'}$. Note that Lemma B.2 shows in the region

$$\{x_1, \ldots, x_{k-1} \geq 2\tau, \tau \leq x_k \leq 2\tau, 0 \leq x_{k+1}, \ldots, x_d \leq \tau\}\,,$$

we have

$$\frac{\partial f(x)}{\partial x_{k+1}} \geq -2\gamma x_{k+1}.$$

Putting this together we obtain the following upper bound, for $t = T_k - T_k^\tau + 1, \ldots, T_k$:

$$x_{k+1}^{(t)} \leq x_{k+1}^{(t')} (1 - 2\eta L)^{(T_k - T_k^\tau - t')} \cdot (1 + 2\eta\gamma)^{t - (T_k - T_k^\tau)} \leq \tau,$$

where the last inequality is because $t - (T_k - T_k^\tau) \leq T_k^\tau \leq \frac{1}{2\eta\gamma}$. This implies $\mathbf{x}^{(t)}$ is in $D_0$. Our proof is complete.

### A.3   Proof for Corollary 4.3

Define $g(\mathbf{x}) = f(\frac{\mathbf{x}-\mathbf{z}}{R})$ to be an affine transformation of $f$, $\nabla g(\mathbf{x}) = \frac{1}{R}\nabla f(\frac{\mathbf{x}-\mathbf{z}}{R})$, and $\nabla^2 g(\mathbf{x}) = \frac{1}{R^2}\nabla^2 f(\frac{\mathbf{x}-\mathbf{z}}{R})$. We see that $\ell_g = \frac{\ell_f}{R^2}$, $\rho_g = \frac{\rho_f}{R^3}$, and $B_g = B_f$, which are $poly(d)$.

Define the mapping $h(x) = \frac{\mathbf{x}-\mathbf{z}}{R}$, and the auxiliary sequence $\mathbf{y}_t = h(\mathbf{x}_t)$. We see that

$$\mathbf{x}^{(t+1)} = \mathbf{x}^{(t)} - \eta \nabla g(\mathbf{x}^{(t)})$$

$$h^{-1}(\mathbf{y}^{(t+1)}) = h^{-1}(\mathbf{y}^{(t)}) - \frac{\eta}{R}\nabla f(\mathbf{y}^{(t)})$$

$$\mathbf{y}^{(t+1)} = h(R\mathbf{y}^{(t)} + z - \frac{\eta}{R}\nabla f(\mathbf{y}^{(t)}))$$

$$= \mathbf{y}^{(t)} - \frac{\eta}{R^2}\nabla f(\mathbf{y}^{(t)}).$$

Thus gradient descent with stepsize $\eta$ on $g$ is equivalent to gradient descent on $f$ with stepsize $\frac{\eta}{R^2}$. The first conclusion follows from noting that with probability $1 - \delta$, the initial point $\mathbf{x}^{(0)}$ lies in $B_\infty(z, R)$, and then applying Theorem 4.1. The second conclusion follows from applying Theorem 2.7 in the same way as in the proof of Theorem 4.1.

## B   Auxiliary Theorems

The following are basic facts from spline theory. See Equation (2.1) and (3.1) of Dougherty et al. [1989]

**Theorem B.1.** *Given data points $y_0 < y_1$, function values $f(y_0)$, $f(y_1)$ and derivatives $f'(y_0)$, $f'(y_1)$ with $f'(y_0) < 0$ the cubic Hermite interpolant is defined by*

$$p(y) = c_0 + c_1 \delta_y + c_2 \delta_y^2 + c_3 \delta_y^3,$$

*where*

$$c_0 = f(y_0), c_1 = f'(y_0)$$

$$c_2 = \frac{3S - f'(y_1) - 2f'(y_0)}{y_1 - y_0}$$

$$c_3 = -\frac{2S - f'(y_1) - f'(y_0)}{(y_1 - y_0)^2}$$

*for $y \in [y_0, y_1]$, $\delta_y = y - y_0$ and slope $S = \frac{f(y_1) - f(y_0)}{y_1 - y_0}$. $p(y)$ satisfies $p(y_0) = f(y_0)$, $p(y_1) = f(y_1)$, $p'(y_0) = f'(y_0)$ and $p'(y_1) = f'(y_1)$. Further, for $f(y_1) < f(y_0) < 0$, if*

$$f'(y_1) \geq \frac{3\,(f(y_1) - f(y_0))}{y_1 - y_0}$$

*then we have $f(y_1) \leq p(y) \leq f(y_0)$ for $y \in [y_0, y_1]$.*

We use these properties of splines to construct the bivariate function $g$ and the univariate function $g_1$ in Section A. The next lemma studies the properties of the connection functions $g(\cdot, \cdot)$ and $g_1(\cdot)$.

**Lemma B.2.** *Define $g(x_i, x_{i+1}) = g_1(x_i) + g_2(x_i)x_{i+1}^2$. There exist polynomial functions $g_1$, $g_2$ and $\nu = -g_1(2\tau) + 4L\tau^2$ such that for any $i = 1, \cdots, d$, for $f_{i,1}$ and $f_{i,2}$ defined in Eq. (6)- (10), $g(x_i, x_{i+1})$ ensures $f_{i,2}$ satisfies, if $x_i = \tau$, then*

$$f_{i,2}(\mathbf{x}) = f_{i,1}(\mathbf{x}),$$
$$\nabla f_{i,2}(\mathbf{x}) = \nabla f_{i,1}(\mathbf{x}),$$
$$\nabla^2 f_{i,2}(\mathbf{x}) = \nabla^2 f_{i,1}(\mathbf{x}),$$

*and if $x_i = 2\tau$ then*

$$f_{i,2}(\mathbf{x}) = f_{i+1,1}(\mathbf{x}),$$
$$\nabla f_{i,2}(\mathbf{x}) = \nabla f_{i+1,1}(\mathbf{x}),$$
$$\nabla^2 f_{i,2}(\mathbf{x}) = \nabla^2 f_{i+1,1}(\mathbf{x}).$$

*Further, $g$ satisfies for $\tau \leq x_i \leq 2\tau$ and $0 \leq x_{i+1} \leq \tau$*

$$-4L\tau \leq \frac{\partial g(x_i, x_{i+1})}{\partial x_i} \leq -2\gamma\tau$$

$$\frac{\partial g(x_i, x_{i+1})}{\partial x_{i+1}} \geq -2\gamma x_{i+1}.$$

*and $g_1$ satisfies for $\tau \leq x_i \leq 2\tau$*

$$-4L\tau \leq \frac{\partial g_1(x_i)}{\partial x_i} \leq -2\gamma\tau.$$

*Proof.* Let us first construct $g_1$. Since we know for a given $i \in [1, \ldots, d]$, if $x_i = \tau$, $\frac{\partial f_{i,1}}{\partial x_i} = -2\gamma\tau$, $\frac{\partial^2 f_{i,1}}{\partial x_i^2} = -2\gamma$ and if $x_i = 2\tau$, $\frac{\partial f_{i+1,1}}{\partial x_i} = -4L\tau$ and $\frac{\partial^2 f_{i+1,1}}{\partial x_i^2} = 2L$. Note for $L > \gamma$, $0 > -2\gamma\tau > -4L\tau$ and $2L > \frac{-4L\tau - (-2\gamma\tau)}{2\tau - \tau}$. Applying Theorem B.1, we know there exists a cubic polynomial $p(x_i)$ such that

$$p(\tau) = -2\gamma\tau \quad \text{and} \quad p(2\tau) = -4L\tau$$
$$p'(\tau) = -2\gamma \quad \text{and} \quad p'(2\tau) = 2L,$$

and $p(x_i) \leq -2\gamma\tau$ for $\tau \leq x_i \leq 2\tau$. Now define

$$g_1(x_i) = \left(\int p\right)(x_i) - \left(\int p\right)(\tau) - \gamma\tau^2.$$

where $\int p$ is the anti-derivative. Note by this definition $g_1$ satisfies the boundary condition at $\tau$. Lastly we choose $\nu = -g_1(2\tau) + 4L\tau^2$. It can be verified that this construction satisfies all the boundary conditions.

Now we consider $x_{i+1}$. Note when if $x_i = \tau$, the only term in $f$ that involves $x_{i+1}$ is $Lx_{i+1}^2$ and when $x_i = 2\tau$, the only term in $f$ that involves $x_{i+1}$ is $-\gamma x_{i+1}^2$. Therefore we can construct $g_2$ directly:

$$g_2(x_i) = -\gamma - \frac{10(L + \gamma)(x_i - 2\tau)^3}{\tau^3} - \frac{15(L + \gamma)(x_i - 2\tau)^4}{\tau^4} - \frac{6(L + \gamma)(x_i - 2\tau)^5}{\tau^5}.$$

Note
$$g_2'(x_i) = -\frac{30(L+\gamma)(x_i - 2\tau)^2(x_i - \tau)^2}{\tau^5}.$$
After some algebra, we can show this function satisfies for $\tau \leq x_i \leq 2\tau$
$$g_2(x_i) \geq -\gamma,$$
$$g_2'(x_i) \leq 0,$$
$$g_2(\tau) = L, \quad g_2(2\tau) = -\gamma$$
$$g_2'(\tau) = g_2'(2\tau) = 0$$
$$g_2''(\tau) = g_2''(2\tau) = 0.$$
Therefore it satisfies the boundary conditions related to $x_{i+1}$. Further note that at the boundary ($x_i = \tau$ or $2\tau$), the derivative and the second derivative are zero, so it will not contribute to the boundary conditions involving $x_i$. Now we can conclude that $g$ and $g_1$ satisfy the requirements of the lemma. $\qquad\square$

We use the following continuous extension theorem which is a sharpened result of the seminal Whitney extension theorem [Whitney, 1934].

**Theorem B.3** (Theorem 1.3 of Chang [2015]). *Suppose $E \subseteq \mathbb{R}^d$. Let the $C^m(E)$ norm of a function $F : E \to \mathbb{R}$ be $\sup\{|\partial^\alpha| : \mathbf{x} \in E, |\alpha| \leq m\}$. If $E$ is a closed subset in $\mathbb{R}^d$, then there exists a linear operator $T : C^m(E) \to C^m(R^d)$ such that if $f \in C^m(E)$ is mapped to $F \in C^m(\mathbb{R}^d)$, then $F|_E = f$ and $F$ has derivatives of all orders on $E^c$. Furthermore, the operator norm $\|T\|_{op}$ is at most $Cd^{5m/2}$, where $C$ depends only on $m$.*