[Reviews · NeurIPS 2017]

Reviewer 1



It has recently been shown that, when all the saddle points of a non-convex function are "strict saddle", then gradient descent with a (reasonable) random initialization converges to a local minimizer with probability one. For a randomly perturbed version of gradient descent, the convergence rate can additionally be shown to be polynomial in the parameters. This article proves that such a convergence rate does not hold for the non-perturbed version: there exists reasonably smooth functions with only strict saddle points, and natural initialization schemes such that gradient descent requires a number of steps that is exponential in the dimension to find a local minimum. I liked this article very much. It answers a very natural question: gradient descent is an extremely classical, and very simple algorithm. Although it is known not to be the fastest one in many situations, it is widely used in practice; we need to understand its convergence rate. The proof is also conceptually simple and elegant, and I found its presentation very clear. My only criticism is that it seems to me that there are a lot of miscalculations in the proof, notably in the proof of Lemma B.2. I absolutely do not think they are a problem for the correctness of the proof, but I would appreciate them to be corrected before the article is published. Minor remarks and typos: - l.91: "postpone in" should be "postpone to" - l.109: ""strict" defined" should be ""strict", which is defined"? - l.115: "the approximation to" should be "approximate"? - Paragraph "initialize far, far away": some condition on h is needed to ensure that x_2 does not suddenly increase when x_1 is between -2 and -1. (I understand that this discussion is informal anyway, but it could be briefly mentioned.) - l.163: "in previous" should be "as in previous" - l.166: why [-1/2;1/2] and not [-1;1]? - l.167: "has" should be "have" - l.173: "from of" should be "from" - l.191: no comma before the closing parenthesis - Corollary 4.3: "The for"? "satisfies" should be "that satisfies"; "same of" should be "same as in" - l.217: "Guassian" - l.218, "where ...": a verb is missing. - Equation (3), second and third line, right hand side: x should be x_1. - l.241: "still satisfy the assumptions" should be "the assumptions are still satisfied" - l.250: "constraint" should be "constrained" - l.251 and 438: "make as" should be "make"? - l.252: remove "both coordinate". - l.261: "(14)and" should be "(14) and" - l.281: "when if" should be "if" - I know that the conjunction "that" can often be omitted after verbs like "know", "note", "imply" ... However, I would personally find the sentences easier to understand if "that" was not omitted when it introduces a complex proposition (lines 274, 388, 391 ...). - l.367: I do not understand "We connnect ... in a spline". - l.376: "where" should be "which" - Page 12: the number of the figure should be written in the caption. - l.397: "above lemma" should be "lemma below" - l.403: remove "for GD" - l.406: "the iterate" should be "the iterates" - Page 13: it seems to me that we need to know that the iterates stay in D_0 in order for the first equations (after line 414) to be valid (which is proved after the equations). - Equation after l.421, second line: I think that the inequality given by Lemma B.2 yields a \geq, not \leq. - Line after this equation: "where the" should be "where for the". - Equation after l.426, second line: a "2" is missing. - I do not think that the definition in Equation (12) is correct (because, with this definition, D_a does not depend on a). - Equation between (15) and (16): shouldn't a distinction also be made on the sign of x_{i+1}? - Step 2: why couldn't we simply define f(x_1,\dots,x_d) by f(x_1,\dots,x_d)=f(|x_1|,\dots,|x_d|)? It seems to me that it is more or less what is done, but written in a more complicated way. Maybe I did not understand this part. - l.460: "agree" should be "agree with" - l.471: maybe also say that there is no noise added when \tau \leq x_k\leq 2\tau? - l.472: I think the assumption should be r \leq \tau/20. - l.478: "same argument" should be "same argument as" - l.496: the comma should be on the previous line. - Theorem B.1: the assumption that f'(y_0)\leq 0 is also needed in order to ensure p(y) \leq f(y_0). - According to my computations, g_1''(\tau) is equal to -4\gamma, while the continuity assumption on the first derivative requires -2\gamma. - l.521: I do not see why -g_1(2\tau)+4L\tau^2 is equal to (17\gamma/6+11 L/2)\tau^2. - I do not see why \nabla f_{i,2} is equal to \nabla f_{i+1,1} (for the derivative along x_i, I do not find the same value). - Theorem B.3: it seems to me that this is an over-simplification of [Chang, 2015, Theorem 1.3] (in particular, but not only, because I think that the notion of C^m functions does not really make sense on generic closed subsets of \R^d).

Reviewer 2



The authors construct examples of smooth bounded non-convex optimization problems where standard gradient descent with random initialization takes exponential time (in dimension) to converge to second order stationary points. In comparison, for such functions, the perturbed version of gradient descent is known converge in polynomial time. The paper provides an important negative result on a significant problem of interest and I vote to accept. That said the examples presented are very special where there gradient path leads though a sequence of d contiguous saddle regions before reaching the local optima. It does not provide much intuition of the general class of problems where GD can or cannot have polynomial time convergence. Writing: Fig 2: What values of tau are used? Also I think contour plots with the saddle regions marked out for construction in (4) and (3) might be easier to interpret. [Edit]: I have read the author response.

Reviewer 3



This paper proves that gradient descent (GD) method can take exponential time to escape strict saddle points. This answers an interesting open question closely related to [Lee et al 16] which showed that GD can escape strict saddle points. As a result, this paper implied that in terms of time complexity, there is indeed a large gap between GD and perturbed GD in the worst case. The answer to the complexity of GD escaping strict saddle points has been much needed for a while, and this paper gave a nice answer to the question. So in my opinion it is a clear acceptance. I have one comment about the initialization. This paper tried hard to go from the “unnatural initialization” to “fairly natural random initialization”. Such an effort is great, and the result is strong and convincing. Nevertheless, choosing initialization method and choosing counter-examples is like a game. This paper first chooses an initialization scheme such as uniform over [0,1], standard Gaussian or an ell_infinity ball as in Corollary 4.3, then choose the counter-example. One immediate question is what if someone chose an initial point on the surface of a large ball containing the local-mins, or a uniform distribution of points in a large cube, say, from -10c to 10c. The answer seems unclear to me. One may argue that in practice people usually fix a simple initialization scheme; but I would say that is because they are already satisfied with that initialization scheme, and if they found the results too bad, experienced practitioners may change the initialization scheme — at least they may try increasing the initialization regime. I don’t expect the authors to analyze the performance of GD for “adaptive random initialization” — this question is just one of many possible questions (like smoothed analysis). A short comment in the paper or the response would be enough.